# Risk of autism spectrum disorder in offspring following paternal use of selective serotonin reuptake inhibitors before conception: a population-based cohort study

Fen Yang,[1,2] Jianping Chen,[1] Mao-Hua Miao,[1] Wei Yuan,[1] Lin Li,[1] Hong Liang,[1] Vera Ehrenstein,[2] Jiong Li[2]

[1]Key Laboratory of Reproduction Regulation of NPFPC, SIPPR, IRD, Fudan University, Shanghai, China
[2]Department of Clinical Epidemiology, Aarhus University Hospital, Aarhus, Denmark

**Correspondence to**
Dr Hong Liang;
lucylhcn@163.com

## ABSTRACT

**Objective** The present study aimed to examine the association between paternal selective serotonin reuptake inhibitor (SSRI) use before conception and the risk of autism spectrum disorder (ASD) in offspring.

**Design** A population-based cohort study.

**Methods** We conducted a cohort study of 669 922 children born from 1998 to 2008, with follow-up throughout 2013. Based on Danish national registers, we linked information on paternal use of SSRIs, ASD diagnosed in children and a range of potential confounders. The children whose fathers used SSRIs during the last 3 months prior to conception were identified as the exposed. Cox regression model was used to estimate the HR for ASD in children.

**Results** Compared with unexposed children, the exposed had a 1.62-fold higher risk of ASD (95% CI 1.33 to 1.96) and the risk attenuated after adjusting for potential confounders, especially fathers' psychiatric conditions (HR=1.43, 95% CI 1.18 to 1.74). When extending the exposure window to 1 year before conception, the increased risk persisted in children of fathers using SSRIs only from the last year until the last 3 months prior to conception (HR=1.54, 95% CI 1.21 to 1.94) but not in children of fathers using SSRIs only during the last 3 months prior to conception (HR=1.17, 95% CI 0.75 to 1.82). We also performed stratified analyses according to paternal history of affective disorders and observed no increased ASD risk among children whose father had affective disorders. Besides, the sibling analysis showed that the ASD risk did not increase among exposed children compared with their unexposed siblings.

**Conclusions** The mildly increased risk of ASD in the offspring associated with paternal SSRI use before conception may be attributable to paternal underlying psychiatric indications related to SSRI use or other unmeasured confounding factors.

## INTRODUCTION

Depression is a common mental health problem that affects approximately 350 million people worldwide.[1] Selective

### Strengths and limitations of this study

► This is the very first study that investigates the association between paternal antidepressant use and autism spectrum disorder (ASD) risk in children.
► A number of potential confounders, including sociodemographic factors as well as parental psychiatric history, can be adjusted for based on the availability of national health registry data.
► Actual use of selective serotonin reuptake inhibitors by fathers during the time period of interest may not be validated.
► Those children who received the diagnosis of ASD from private psychiatrists or remained undiagnosed at the end of the follow-up may not be identified as ASD case.

serotonin reuptake inhibitors (SSRI) are commonly prescribed for the treatment of depression and other anxiety-related disorders. However, an increasing body of evidence suggests that prenatal exposure to SSRIs is associated with adverse obstetrical and neonatal outcomes.[2–5] In addition, recent studies have indicated a possible link between prenatal SSRIs exposure and neurobehavioral problems in children,[2 6–10] which may be due to the biological mechanisms (ie, dysfunctional serotonin signalling).[11] However, several studies have suggested that the previously observed association between prenatal SSRIs exposure and neurodevelopmental diseases might be due to confounding by related indications.[12–15]

Autism spectrum disorder (ASD) is one of the major neurodevelopmental disorders characterised by impairments in reciprocal social interaction, communication and repetitive or stereotypic behavior.[16] The estimated prevalence of ASD has dramatically

**BMJ**

increased from less than 1 in 2000 children in 1960s to approximately 1 in 145 children today.[17 18] Both genetic and environmental risk factors may contribute to ASD.[19] Recently, maternal use of SSRIs during pregnancy has been reported to be associated with an increased risk of ASD in children.[10 20–24]

If maternal SSRIs exposure during pregnancy plays a role in fetal, even in child's, development, an extended effect of earlier exposure on gamete, particularly human sperm, is biologically plausible. Therefore, the potential role of paternal SSRI use in ASD should also be taken into consideration. Studies in the Nordic countries show that one-third of the fathers used prescription drugs during the last 6 months prior to conception,[25] and approximately 1.4% of fathers were dispensed SSRIs during the last 3 months prior to conception.[26 27] Although the information regarding the potential effects of SSRI use by fathers is limited, the hypothesis of an increased risk of adverse pregnancy or neonatal outcome associated with paternal drug exposure is not new. Several human studies have indicated the adverse effects of paternal SSRI use, including impaired semen quality and abnormal sperm DNA fragmentation, which has been reported to be associated with diminished fertility, adverse pregnancy outcomes (like pregnancy loss) and an increased risk of childhood disease.[28–30]

Little is known on whether paternal SSRI use before conception contributes to the risk of ASD in offspring. We conducted a population-based cohort study to examine the association between paternal SSRI use during the last 3 months prior to conception and risk of ASD in offspring, using data from national Danish health registries.

## METHODS
### Study population
This study was based on several national registers in Denmark. Each Danish resident is assigned a unique personal identification number (a 10-digit civil registration system number used in all registries), which enables accurate linkage of national registries at the individual level.[31] The Danish Medical Birth Registry (DMBR) contains records of all deliveries in Denmark since 1973 and includes information about gestational age at birth from 1978.[32] Using the DMBR data, we identified a cohort of all singletons born alive in Denmark during the period from 1 January 1998 to 31 December 2008 (n=687 580). We excluded children without linkage to their father (n=16 955) or their mother (n=74), with missing parity (n=208), and with missing or extreme values of gestational age (≤23 weeks or ≥45 weeks, n=421). A total of 669 922 children were included in the analysis.

### Data on SSRI use
Information on SSRI use was drawn from the Danish National Prescription Registry.[33] Since 1995, this registry has recorded all redeemed prescriptions in Denmark with the following information: the civil registration number

of the patient, the dispensing date, the medication code (the WHO Anatomical Therapeutic Chemical (ATC) classification system), the number of packages prescribed and the number of dose units in package. SSRI use was identified based on ATC codes: fluoxetine (N06AB03), citalopram (N06AB04), paroxetine (N06AB05), sertraline (N06AB06), fluvoxamine (N06AB08) and escitalopram (N06AB10). It is estimated that spermatogenesis takes approximately 74 days,[34] therefore we chose the last 3 months prior to conception as the exposure window to cover the susceptible time period. A child was considered exposed if the dispensing date fell within the specified exposure window or the number of days for which the SSRI medication was supplied overlapped any portion of the exposure window. Children born to fathers who had no prescriptions for SSRIs and no supply overlap during the entire exposure window were considered unexposed. The date of conception was estimated by subtracting gestational age from the date of delivery. Data on paternal SSRI use during the last 2 years prior to the conception were extracted for further analyses. We also retrieved the information about maternal SSRI use during the pregnancy.

### Autism spectrum disorders
ASDs in children were identified by using the Danish Psychiatric Central Research Register (DPCRR) and the Danish National Patient Register (DNPR). The DPCRR contains diagnostic information on every admission from psychiatric hospitals and psychiatric wards in general hospitals in Denmark since 1969, and includes data on all outpatient visits and emergency room contacts since 1995.[35] The DNPR has collected data on all inpatients from all somatic hospitals in Denmark since 1977 and outpatients from 1995.[36] The combined data from the two registries were used to identify all children diagnosed with ASD. During the study period, the diagnosis of ASD was based on the International Classification of Diseases, 10th version (ICD-10) codes of F84.0 (childhood autism), F84.1 (atypical autism), F84.5 (Asperger syndrome), F84.8 (other pervasive developmental disorders) and F84.9 (unspecified pervasive developmental disorders). The quality of the childhood autism diagnosis has been validated and the diagnoses could be verified in 94% of the children with a record in the DPCRR.[37] Children were followed up from birth until first diagnosis of ASD, death, emigration, or 31 December 2013, whichever came first.

### Covariates
Using the Danish nationwide health registers, we retrieved data on characteristics that may be associated with offspring ASD or paternal SSRI use. For each child, we obtained information on calendar year of birth, gender of the child, birth weight, Apgar score at 5 min, gestational age, maternal parity, maternal age at child birth, paternal age at child birth and maternal smoking status during pregnancy from the DMBR. Parents' psychiatric history prior to birth of the index child was obtained from

DPCRR by ICD-8 codes 290–315 from 1977 to 1993 and ICD-10 codes F00–F99 from 1994 onwards. Furthermore, we identified parents diagnosed with affective disorders before birth of child (specifically, ICD-8 codes 296.09, 296.19, 296.29, 296.39, 296.99, 298.09, 298.19, 300.49 and 301.19, and ICD-10 codes F30–F34 and F38–F39) using the DPCRR.

## Statistical analysis

Cox proportional hazards regression models (using child's calendar age in years as the underlying timescale) were used to estimate the HRs and 95% CIs of ASD in children following exposure to paternal SSRI use before conception. Observations were censored if the child died or emigrated before the end of follow-up.

We adjusted for variables as follows: the calendar year of birth (1998–2000, 2001–2003, 2004–2006, 2007–2008), gender of the child (boy, girl), parity (1, 2, ≥3), parental age at child birth (≤25, 26–30, 31–35 and >35 years), maternal smoking status during pregnancy (yes or no), maternal history of psychiatric disorders before birth of child (yes or no) and maternal antidepressant (AD) use during pregnancy (yes or no) in model 1. We additionally adjusted for paternal history of psychiatric disorders before birth of child (yes or no) in model 2. Models were also run with the exclusion of those children with missing data for covariates. The proportional hazard assumption was evaluated for all variables included in the adjusted model by comparing estimated log-minus-log survival curves.

To distinguish the direct effects of SSRI use from the effects of the indication of SSRI use, we extended the exposure window to 1 year before conception. We then recategorised the exposed children into three subgroups: children of fathers who used SSRIs: (1) only from the last year until the last 3 months prior to conception (former users); (2) only during the last 3 months prior to conception (current short-term users, hereinafter referred to as 'current users'); (3) both before and during the last 3 months prior to conception (both former and current users). The reference group consisted of those children born to fathers who never used SSRI medication through the last year before conception.

The stratified analysis was performed to examine whether the association between paternal SSRI use and ASD in children differed by gender. We also restricted the analyses to children whose mothers neither received AD medication during pregnancy nor had affective disorder before birth of child. To further distinguish the effect of SSRI medication from that of the main indication (ie, affective disorders) for SSRI treatment, we performed stratified analyses according to paternal history of affective disorder before birth of the index child. As for those children born to fathers with affective disorders, the ASD risk we examined could be solely attributable to paternal SSRI use since both the exposed children and the reference children were with paternal affective disorders.

To control for unmeasured family-related confounding factors (such as genetic liability for neuropsychiatric conditions and early postnatal environmental influences), we conducted sibling-matched analyses by only including the families with exposure-discordant siblings, in which there was at least one child with paternal SSRI preconception exposure and one child without exposure. Using the stratified Cox proportional hazards regression with a separate stratum for each family, we estimated the association between paternal SSRI use before conception and ASD in matched sets of exposure-discordant siblings. In the stratified Cox regression model, we also adjusted for those covariates that varied among siblings with the same father (including parental age at conception, maternal parity, smoking and maternal AD use during pregnancy).

All analyses were performed using SAS V.9.1 (SAS Institute).

## RESULTS

Among the 669 922 singletons included in the study, 6870 (1.03%) children were born to fathers who had redeemed a prescription for SSRIs during the last 3 months prior to conception. During the study period, a total of 7577 children were diagnosed with ASD. The median age at diagnosis of ASD was 6.77 years (IQR: 4.84–9.48 years). Characteristics of the study population are shown in table 1. Compared with the unexposed group, there was a higher proportion of exposed children born in later calendar years. Fathers in exposed group were more likely to be older at child birth and to have a history of psychiatric disorder (including affective disorder) before birth of child. Mothers of exposed children were characterised as having higher parity, being more likely to be older at child birth, to use ADs during pregnancy and to have a history of psychiatric disorder before birth of child.

Among 61 555 person-years of follow-up, we identified 104 cases of ASD in children born to fathers who used SSRIs during the last 3 months prior to conception (incidence rate: 169 per 100 000 person-years). This incidence rate corresponded to a 62% increased risk of ASD compared with the unexposed group (table 2). The adjusted HR (aHR) of ASD was 1.54 (95% CI 1.27 to 1.88) after adjusting for potential confounders in model 1. After paternal psychiatric history was further adjusted for in model 2, the estimate was 1.43 (95% CI 1.18 to 1.74). When extending the exposure window to the last 1 year prior to conception, after the full adjustment, the HR for ASD in children of fathers who were former users only and who were both former and current users was 1.54 (95% CI 1.21 to 1.94) and 1.32 (95% CI 1.02 to 1.72), respectively. However, the increased risk decreased and became non-significant among children of fathers who were only current users (aHR=1.17, 95% CI 0.75 to 1.82, model 2, table 2).

The risk estimates of ASD were similar for both boys and girls, regardless of the exposure (online supplementary

| Table 1 | Baseline characteristics of the study population | |
|---|---|---|
| **Characteristic** | **Paternal SSRI use during the last 3 months prior to conception (n=6870)** | **No paternal SSRI use during the last 3 months prior to conception (n=663 052)** |
| Calendar year of birth, n (%) | | |
| 1998–2000 | 1021 (14.9) | 185 784 (28.0) |
| 2001–2003 | 1526 (22.2) | 179 952 (27.1) |
| 2004–2006 | 2305 (33.5) | 178 664 (27.0) |
| 2007–2008 | 2018 (29.4) | 118 652 (17.9) |
| Gender, n (%) | | |
| Boy | 3549 (51.7) | 340 194 (51.3) |
| Girl | 3321 (48.3) | 322 858 (48.7) |
| Birth weight (g), n (%) | | |
| <2500 | 278 (4.1) | 22 315 (3.4) |
| 2500–3250 | 1729 (25.2) | 154 494 (23.3) |
| 3250–4000 | 3582 (52.1) | 348 853 (52.6) |
| 4000–8000 | 1243 (18.1) | 132 343 (19.9) |
| Unknown | 38 (0.5) | 5047 (0.8) |
| Parity, n (%) | | |
| 1 | 2693 (39.2) | 282 895 (42.7) |
| 2 | 2433 (35.4) | 250 496 (37.8) |
| ≥3 | 1744 (25.4) | 129 661 (19.5) |
| Preterm birth, n (%) (<37 weeks) | | |
| No | 6489 (94.4) | 630 851 (95.1) |
| Yes | 381 (5.6) | 32 201 (4.9) |
| Apgar score at 5 min, n (%) | | |
| 0–7 | 91 (1.3) | 8224 (1.2) |
| 8–9 | 442 (6.4) | 39 941 (6.0) |
| 10 | 6269 (91.3) | 607 634 (91.7) |
| Unknown | 68 (1.0) | 7253 (1.1) |
| Maternal age at child birth (years), n (%) | | |
| ≤25 | 958 (13.9) | 99 273 (15.0) |
| 26–30 | 2238 (32.6) | 244 419 (36.9) |
| 31–35 | 2386 (34.7) | 225 610 (34.0) |
| >35 | 1288 (18.8) | 93 750 (14.1) |
| Paternal age at child birth (years), n (%) | | |
| ≤25 | 364 (5.3) | 47 490 (7.2) |
| 26–30 | 1434 (20.9) | 180 612 (27.2) |
| 31–35 | 2395 (34.8) | 242 702 (36.6) |
| >35 | 2677 (39.0) | 192 248 (29.0) |
| Maternal smoking status*, n (%) | | |
| No | 4326 (63.0) | 477 574 (72.0) |
| Yes | 1334 (19.4) | 106 233 (16.0) |
| Unknown | 1210 (17.6) | 79 245 (12.0) |
| | | Continued |

| Table 1 | Continued | |
|---|---|---|
| **Characteristic** | **Paternal SSRI use during the last 3 months prior to conception (n=6870)** | **No paternal SSRI use during the last 3 months prior to conception (n=663 052)** |
| Maternal AD use during pregnancy, n (%) | | |
| No | 6548 (94.0) | 653 496 (98.6) |
| Yes | 412 (6.0) | 9556 (1.4) |
| Maternal history of affective disorder, n (%) | | |
| No | 6607 (96.2) | 654 234 (98.7) |
| Yes | 263 (3.8) | 8818 (1.3) |
| Paternal history of affective disorder, n (%) | | |
| No | 6099 (88.8) | 659 639 (99.5) |
| Yes | 771 (11.2) | 3413 (0.5) |
| Maternal history of psychiatric disorder, n (%) | | |
| No | 6175 (89.9) | 626 840 (94.5) |
| Yes | 695 (10.1) | 36 212 (5.5) |
| Paternal history of psychiatric disorder, n (%) | | |
| No | 5092 (74.1) | 630 990 (95.2) |
| Yes | 1178 (25.9) | 32 062 (4.8) |

*Maternal smoking information was missing among those children born from 2007 to 2008.
AD, antidepressant drug; n, number; SSRI, selective serotonin reuptake inhibitor.

table S1). When we restricted analyses to children whose mothers neither used SSRIs during pregnancy nor had affective disorder before birth of child, the results did not change essentially (online supplementary table S2).

When the analyses were stratified by paternal affective disorder before birth of the index child, in children born to fathers with a history of affective disorder, there was no association between paternal SSRI use before conception and ASD in the offspring (model 2: aHR=1.10, 95% CI 0.61 to 1.98). For children whose father had no affective disorder, the patterns of associations remained similar to those of the main analyses (table 3).

In the sibling analysis, we identified 5479 families with more than one child and with at least one child with paternal SSRI use before conception (table 4). The risk of ASD in exposed children was decreased when compared with their unexposed siblings (aHR=0.74, 95% CI 0.34 to 1.59), although with wider CI.

## DISCUSSION

In this large population-based cohort study, we observed an increased risk of ASD in the offspring following paternal use of SSRIs during the last 3 months prior to conception. However, the risk attenuated after adjusting for a number of potential confounders, especially fathers'

**Table 2** Association between paternal SSRI use before conception and ASD in offspring

| Paternal SSRI use before conception | Offspring with ASD, n | Follow-up number of person-years | HR (95% CI) | | |
| --- | --- | --- | --- | --- | --- |
| | | | Crude | Model 1* | Model 2† |
| No use during the last 3 months prior to conception | 7473 | 6 765 205 | Ref | Ref | Ref |
| Use during the last 3 months prior to conception | 104 | 61 555 | 1.62 (1.33 to 1.96) | 1.54 (1.27 to 1.88) | 1.43 (1.18 to 1.74) |
| Subanalysis: paternal SSRI use during the last 1 year before conception | | | | | |
| No use during the last 1 year prior to conception | 7429 | 6 736 654 | Ref | Ref | Ref |
| Use only from the last 1 year to the last 3 months prior to conception | 71 | 39 422 | 1.71 (1.35 to 2.16) | 1.66 (1.31 to 2.09) | 1.54 (1.21 to 1.94) |
| Use only during the last 3 months prior to conception | 20 | 14 738 | 1.29 (0.83 to 2.00) | 1.24 (0.80 to 1.93) | 1.17 (0.75 to 1.82) |
| Use both before and during the last 3 months prior to conception | 57 | 35 946 | 1.52 (1.17 to 1.97) | 1.43 (1.10 to 1.86) | 1.32 (1.02 to 1.72) |

*Adjusted for calendar year of birth, sex, parity, mother age, father age, maternal smoking, mother psychiatric history, maternal AD use during pregnancy.
†Model 1 further adjusted for father psychiatric history.
AD, antidepressant drug; ASD, autism spectrum disorder; n, number; SSRI, selective serotonin reuptake inhibitor.

psychiatric conditions. When extending the exposure window to 1 year before conception, the ASD risk persisted among children born to former users but not current users. In addition, among children born to father with affective disorder, no association was observed. Finally, we performed a sibling analysis which allowed for better control of unmeasured familial confounding and the decreased ASD risk was found among exposed children rather than their unexposed siblings. Taken as a whole, our results did not support that paternal SSRI use before conception could increase the risk of ASD in children.

Recently, concerns have been raised regarding the risk of ASD in the offspring associated with prenatal exposure to SSRIs. Four previous human studies have suggested that in utero exposure to ADs would increase the risk of ASD in children,[21–24] although another two studies using the Danish registers have reported no significant association.[38 39] Basic neurobiological studies have showed that prenatal SSRI administration may be part of a causal pathway to ASD by operating directly on the developing brain.[40 41] Considering the detrimental effect of SSRIs on sperm,[28 29 42] paternal SSRI use before conception may also cause adverse effects on pregnancy outcomes. However, until now, only one study has suggested that SSRI administration to male rats could induce deterioration in fertility and fetal outcomes (including weight gain, organ weights and feed consumption).[43] To our knowledge, our study is the first to investigate the link between paternal AD use before conception and ASD risk in children. Considering the limited number of ASD cases and the challenges posed by confounding, further studies are needed to corroborate the findings.

In the present study, the increased risk of ASD observed in former users but not in current users implied that indications for paternal SSRI use may account for the observed association between paternal SSRI use and ASD in children, which was supported by previous studies suggesting that parental depression and other psychiatric disorders might be associated with the risk of autism in children.[44–46] Therefore, similar to those studies which focused on the effect of maternal AD use during pregnancy, confounding by indication poses the main challenge in our study. We adopted several analytical strategies to account for such confounding by indication: (1) regression adjustment for paternal psychiatric disorders; (2) negative controls (ie, former user analyses); (3) stratified analyses according to paternal history of affective disorders; and (4) sibling analyses. The results of these analyses suggested that paternal psychiatric illness rather than SSRI exposure might be associated with ASD liability. It was worth noting that, in children born to fathers without affective disorders, there was a significantly increased risk associated with paternal SSRI use. Most fathers might receive SSRI treatment from their general practitioner and were therefore not registered with a diagnosis of affective disorders in the hospital system. Therefore, it was possible that the increased risk associated with prenatal SSRI use was partly confounded by paternal affective disorders diagnosed outside a hospital department for which we were not able to adjust. In addition, other psychiatric diseases related to SSRI use might also contribute to the observed association.

Our study has several methodological strengths. One strength was that the linkage of several nationwide

**Table 3** Association between paternal SSRI use before conception and ASD in offspring born to fathers diagnosed with or without affective disorder

| Paternal SSRI use before conception | Offspring with ASD, n | Follow-up number of person-years | HR (95% CI) | | |
| --- | --- | --- | --- | --- | --- |
| | | | Crude | Model 1* | Model 2† |
| **Fathers with affective disorder** | | | | | |
| No use during the last 3 months prior to conception | 58 | 30 908 | Ref | Ref | Ref |
| Use during the last 3 months prior to conception | 14 | 6625 | 1.17 (0.65 to 2.09) | 1.10 (0.61 to 1.98) | 1.10 (0.61 to 1.98) |
| Subanalysis: paternal SSRI use during the last 1 year before conception | | | | | |
| No use during the last 1 year prior to conception | 53 | 28 033 | Ref | Ref | Ref |
| Use only from the last 1 year to the last 3 months prior to conception | 8 | 4082 | 1.07 (0.51 to 2.25) | 1.08 (0.51 to 2.30) | 1.08 (0.51 to 2.30) |
| Use only during the last 3 months prior to conception | 2 | 1164 | 0.91 (0.22 to 3.75) | 0.89 (0.21 to 3.75) | 0.89 (0.21 to 3.75) |
| Use both before and during the last 3 months prior to conception | 9 | 4254 | 1.16 (0.57 to 2.35) | 1.10 (0.55 to 2.22) | 1.10 (0.55 to 2.22) |
| **Fathers without affective disorder** | | | | | |
| No use during the last 3 months prior to conception | 7415 | 6 752 899 | Ref | Ref | Ref |
| Use during the last 3 months prior to conception | 90 | 36 329 | 1.57 (1.27 to 1.93) | 1.52 (1.23 to 1.87) | 1.45 (1.18 to 1.79) |
| Subanalysis: paternal SSRI use during the last 1 year before conception | | | | | |
| No use during the last 1 year prior to conception | 7376 | 6 714 926 | Ref | Ref | Ref |
| Use only from the last 1 year to the last 3 months prior to conception | 63 | 37 973 | 1.69 (1.32 to 2.17) | 1.64 (1.28 to 2.11) | 1.57 (1.23 to 2.02) |
| Use only during the last 3 months prior to conception | 18 | 7354 | 1.26 (0.79 to 2.00) | 1.23 (0.77 to 1.95) | 1.19 (0.75 to 1.88) |
| Use both before and during the last 3 months prior to conception | 48 | 28 974 | 1.45 (1.09 to 1.93) | 1.38 (1.04 to 1.84) | 1.32 (0.99 to 1.76) |

*Adjusted for calendar year of birth, parity, mother age, father age, maternal smoking, mother psychiatric history, maternal AD use during pregnancy.
†Model 1 further adjusted for father psychiatric history.
AD, antidepressant drug; ASD, autism spectrum disorder; n, number; SSRI, selective serotonin reuptake inhibitor.

health registries in Denmark enabled us to conduct a large cohort study with virtually complete follow-up. The definition on exposure to SSRIs was based on a national registry, which eliminated the risk of recall bias caused by self-report. Another strength was that the information on ASD diagnosis was obtained independently of exposure measurement, which could also mitigate the information bias. Furthermore, the availability of health registry data enabled us to adjust for a number of potential confounders including sociodemographic factors as well as parental psychiatric history. Besides, we have taken the potential effect of maternal AD use during pregnancy as well as maternal mental disease into consideration. To remove the confounding factors attributing to mothers, we adjusted the maternal SSRI use in regression model, and also restricted the analyses to children whose mothers neither received AD medication during pregnancy nor had affective disorders before child birth.

 Yang F, *et al. BMJ Open* 2017;**7**:e016368. doi:10.1136/bmjopen-2017-016368

**Table 4** Paternal SSRI use before pregnancy and ASD in exposed and unexposed siblings from 5479 families

| Paternal SSRI use before conception | Offspring, n | Offspring with ASD, n | HR (95% CI) | |
|---|---|---|---|---|
| | | | Crude | Model 1* |
| No use during the last 3 months prior to conception | 2792 | 45 | Ref | Ref |
| Use during the last 3 months prior to conception | 2687 | 23 | 0.53 (0.31 to 0.91) | 0.74 (0.34 to 1.59) |

*Adjusted for sex, parity, mother age, father age, maternal smoking, maternal AD use during pregnancy.
AD, antidepressant drug; ASD, autism spectrum disorder; n, number; SSRI, selective serotonin reuptake inhibitor.

Limitations need to be considered when interpreting the results of our study. First, we were unable to validate actual use of SSRIs by fathers during the time period of interest because we relied on medical records of dispensed prescriptions. This may lead to misclassification of exposure status because some people may not take the medication or may take it later. Nevertheless, the misclassification was most likely non-differential, which could bias the association towards null. Besides, some patients may receive SSRI treatments during inpatient admissions which are not included in the prescription registry. We expect this problem to be minor since those inpatients usually have severe psychiatric disorders and are more likely to continue treatment after discharge. Second, the 3-month cut-off point was set based on the fact that spermatogenesis takes approximately 70–90 days in humans. However, it may be possible that SSRI drugs induce sperm damage at the very primitive stage. However, the results did not change markedly after extending the putative exposure period to the last 6 months prior to conception (data not shown). Third, ASD in children was ascertained through the DNPR and the DPCRR, which did not include those children who received the diagnosis of ASD from private psychiatrists or remained undiagnosed at the end of the follow-up. However, the prevalence of ASD in our cohort was 1.13%, which was similar to that reported in the USA during the study period (1.14%).[47] Hence, the bias introduced by case identification was expected to be minimal. Fourth, the age of ASD diagnosis which was used as time event in Cox regression models might be affected by external and extraneous factors. If these factors are differentially distributed in exposed and unexposed groups, the actual associations may be biased. We have adjusted for some factors which may influence age of diagnosis to reduce the bias to some extent. However, we could not rule out the confounding effects of unmeasured factors, which is a limitation of the study.

## CONCLUSIONS

In the study, paternal SSRI use before conception was associated with an increased risk of ASD in the offspring, especially in the former users who took SSRIs over the longer term. However, null association was observed in exposed children with paternal affective disorders, and similar ASD risk was observed among exposed and unexposed siblings, which implicates that paternal underlying indications related to SSRI use or other unmeasured confounding factors may explain the increased risk.

**Contributors** HL had full access to all the data in the study and took responsibility for the integrity of the data and the accuracy of the data analysis. HL, WY and JL conceptualised and designed the study, and MHM and VE helped with its development. FY, LL and JPC conducted the statistical analysis. HL, JPC and FY interpreted the results and FY drafted the initial manuscript. All authors reviewed the manuscript and approved the final version as submitted.

**Funding** The study was supported by the National Natural Science Foundation of China (81428011), the National Key Research and Development Program of China (2016YFC1000505), the European Research Council (ERC-2010-StG-260242-PROGEURO), the Nordic Cancer Union (176673, 186200), the Danish Council for Independent Research (DFF-6110-00019), the Karen Elise Jensens Fond (2016), and the Program for Clinical Research Infrastructure (PROCRIN) established by the Lundbeck Foundation and the Novo Nordisk Foundation.

**Disclaimer** The funding sources had no role in the design and conduct of the study; collection, management, analysis and interpretation of the data; preparation, review, or approval of the manuscript; and decision to submit the manuscript for publication. The researchers were independent from the funders.

**Competing interests** None declared.

**Ethics approval** The study was based on secondary data. No individuals were approached as a result of the study, nor did we access any other data from the participants. This study was approved by the Danish Data Protection Agency (Document No. 2013-41-2569). All procedures performed in the study involving human participants were in accordance with the ethical standards of the institutional and/or national research committee and with the 1964 Declaration of Helsinki and its later amendments or comparable ethical standards. Approval by an institutional review board and informed consent are not required for registry-based research in Denmark.

**Provenance and peer review** Not commissioned; externally peer reviewed.

**Data sharing statement** No additional data are available.

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
