## [Reviewer comments · BMJ Open]

ARTICLE DETAILS

TITLE (PROVISIONAL)	Risk of autism spectrum disorder in offspring following paternal use of selective serotonin reuptake inhibitors before conception: a population-based cohort study
AUTHORS	Yang, Fen; Chen, Jianping; Miao, Mao-Hua; Yuan, Wei; Li, Lin; Liang, Hong ; Ehrenstein, Vera; Li, Jiong

VERSION 1 – REVIEW

REVIEWER	Professor David Healy North Wales Department of Psychological Medicine Bangor University Wales UK
REVIEW RETURNED	22-Mar-2017

GENERAL COMMENTS	ABSTRACT - concludes 'findings suggested no substantial increase in the risk of ASD in the offspring attributable to paternal SSRIs use before conception'. However the results clearly do show an increased risk of ASD in offspring to fathers who have previously taken an SSRI and this risk is higher in those who have taken it within the previous year before conception (former users – 1 year to 3 months) and those who have more long term use of the antidepressant (1 year to current use). INTRODUCTION – could include more up to date research. Authors only cite 5 papers indicating possible link between prenatal SSRI exposure and neurobehavioural problems in children: 1. Oberlander et al 2007 2. Nulman et al 2012 – (NB Nulman et al paper actually concluded that maternal depression was the risk factor not SSRI exposure). 3. Gidaya et al 2014 4. Croen et al 2011 5. Rai et al 2013 There are a number of other papers missing (see HEALY et al 2016 – for systematic review. See below for full reference). METHODS Data on SSRI Use: Authors identified SSRI use based on WHO ATC codes for 6 different drugs (fluoxetine, citalopram, paroxetine, sertraline, fluvoxamine, escitalopram) – authors could give a breakdown analysis looking at the effects of these different drugs individually if the numbers in each group allowed.
--

There are other similar drugs that could also be included in the analysis. (i.e Duloxetine, and venlafaxine). The analysis looks at children exposed – that is those whose fathers who took an SSRI within the 3 month window – and those who did not take an SSRI within the 3 month window (but may have taken an SSRI at some point in the previous 1 year or more). Ideally the main analysis should be a comparison between those exposed and those children from fathers who have NEVER taken an SSRI – to control for possible long term/ protracted effects of the SSRI on sperm production.

RESULTS

CRUDE – use of an SSRI in 3 months prior to conception – 1.62 (1.33-1.96) – and gender analysis seems to show this is mainly effecting the boys, at least in the short term initially – Boys: 1.42 (0.89-2.26) vs Girls: 0.67 (0.17-2.69).

Former vs LT users vs ST users (adjusted) – 1.54 (1.21-1.94) vs 1.32 (1.02-1.72) vs 1.17 (0.75-1.82) – these are the results after adjustment for 9 potential confounders.

DISCUSSION

Authors open their Discussion with ‘we observed an increased crude risk of ASD in the offspring following paternal use of SSRIs during the last 3 months prior to conception’ however claim that following adjustment for confounders that this risk disappears, which is misleading. Even after the adjustment of NINE ‘possible’ confounders the results still show an increased risk.

‘Risk persists among former users but not current’ – results showed an increase risk for both but higher in former. Also labelling ‘current’ as ‘current SHORT TERM’ user might be more accurate.

Authors state ‘When we looked into those children without paternal affective disorder before birth of children, similar patterns of association to the main analyses were found. Thus, in addition to affective disorders, other indications related to SSRIs use may also contribute to the observed associations.’

The analysis showed a higher risk in those children born to fathers without a mood disorder (1.57: 1.27-1.93) compared to those with a diagnosis of a mood disorder: 1.17: 0.65-2.09). This does not support the argument that a diagnosis of a mood disorder itself is a significant risk factor, at least in men.

OVERALL CONCLUSION:

‘Our evidence does not support that paternal SSRI use before conception increases the risk of ASD in the offspring, but implies that paternal underlying indications related to SSRI use, or other unmeasured confounding factors may play a role.’

The results show that there IS an increased risk of ASD associated with paternal use of an SSRI and this risk seems more pronounced even after cessation of the drug and over the longer term. The results also show that a diagnosis of an affective disorder does not increase the risk of ASD in offspring but that the risks seem higher in offspring from patients with other indications requiring the use of an SSRI.

It would be helpful if these results were broken out by drug where possible. The assumption is that all serotonin reuptake inhibiting drugs produce changes in sperm count and function but any evidence on this would be useful for formulating further investigations.

The pattern points to effects on sperm count and function rather than the availability of the drug in seminal fluid.

	REFERENCES Healy, D., Le Noury, J.C & Mangin, D. (2016) Links between serotonin reuptake inhibition during pregnancy & "Autistic Spectrum Disorders": a systematic review of epidemiological and physiological evidence. International Journal of Risk and Safety in Medicine. 28, 125-141. doi: 10.3233/JRS-160726.
--	--

REVIEWER	SW Leung University of Macau, Macao, China
REVIEW RETURNED	17-Apr-2017

GENERAL COMMENTS	The manuscript reported non-significant/negative results in finding any role of paternal SSRI use in developing ASD in children. It cited only references 11-13 to mention about the seemingly [still not enough to confirm] significant roles of maternal SSRI use in developing ASD. It seems that the current study design was too limited to analyse and contrast the possible roles of paternal and maternal SSRI use. Without the data contrast (positive in maternal roles and negative in paternal roles) in the same study, it would not be convincing to me that the study is highly significant in design and sensitive in methods.
---

VERSION 1 – AUTHOR RESPONSE

Reviewer: 1

1. ABSTRACT - concludes 'findings suggested no substantial increase in the risk of ASD in the offspring attributable to paternal SSRIs use before conception'.

However the results clearly do show an increased risk of ASD in offspring to fathers who have previously taken an SSRI and this risk is higher in those who have taken it within the previous year before conception (former users – 1 year to 3 months) and those who have more long term use of the antidepressant (1 year to current use).

Response: Thank you for your comments. As the results shown in Table 2, we did find a significantly increased risk of autism spectrum disorder (ASD) in association with the paternal use of selective serotonin reuptake inhibitors (SSRIs) before conception. However, this observed association might be attributable to confounding by indication for SSRIs treatment. Therefore, we performed several analytic strategies to account for such confounding by indication, and one of them was negative controls (i.e., former-users analyses, which indicated the effects of underlying indications). In this way, the association related to paternal prenatal SSRIs exposure (current users who used SSRIs only during the last 3 months prior to conception) attenuated and became statistically non-significant. That was why we concluded 'no substantial increase in the risk of ASD in the offspring attributable to paternal SSRIs use before conception'. Accordingly we have revised our conclusions in the ABSTRACT section.

2. INTRODUCTION – could include more up to date research. Authors only cite 5 papers indicating possible link between prenatal SSRI exposure and neurobehavioural problems in children. There are a number of other papers missing (see HEALY et al 2016 – for systematic review. See below for full reference).

Response: Thank you for your suggestions, and we have added several latest references in the INTRODUCTION section. In addition, we have added some latest studies published in 2017 which suggested that the observed link between prenatal SSRIs use and neurodevelopmental diseases may be attributable to the underlying indications.

3. METHODS-Data on SSRI Use: Authors identified SSRI use based on WHO ATC codes for 6 different drugs (fluoxetine, citalopram, paroxetine, sertraline, fluvoxamine, escitalopram) – authors could give a breakdown analysis looking at the effects of these different drugs individually if the numbers in each group allowed. There are other similar drugs that could also be included in the analysis. (i.e Duloxetine, and venlafaxine).

Response: Thank you for your suggestions. As for fluvoxamine and escitalopram, the cases in the exposed group (1 ASD case and 6 ASD cases, respectively) were too few to run analyses. We have examined the effects of the other subtypes of SSRIs individually, and patterns of the associations were essentially unchanged (Please find the related results in supplementary tables entitled as 'supplementary tables for the response to the reviewer': fluoxetine-table s3, paroxetine-table s4, citalopram-table s5, sertraline-table s6). As for the effect of other antidepressants drug that was similar to SSRIs, like Duloxetine and Venlafaxine, the cases (0 ASD case and 9 ASD cases, respectively) were also too rare to run analyses.

The analysis looks at children exposed – that is those whose fathers who took an SSRI within the 3 month window – and those who did not take an SSRI within the 3 month window (but may have taken an SSRI at some point in the previous 1 year or more). Ideally the main analysis should be a comparison between those exposed and those children from fathers who have NEVER taken an SSRI – to control for possible long term/ protracted effects of the SSRI on sperm production.

Response: Thank you for your suggestion. We retrieved the information on SSRIs use from the Danish National Prescription Registry (DNPR), which has recorded redeemed prescription since 1995. In addition, we recruited all children born alive since Jan 1998 as study population to ensure a relatively large sample size. Thus, we only can retrieve fathers' SSRIs use information 2 years before conception. Antidepressants (including SSRIs) are, however, used for long-term treatment and the related psychiatric disorder is usually severe enough and needs continuous treatment. Hence, those men who did not take SSRIs during the last 2 years before conception might represent most of those who never take SSRIs. Considering that, we have re-defined those children whose father did not take SSRIs during the last 2 years before conception as the reference group, and have run the analyses again. We found the patterns of the associations were essentially unchanged ('supplementary tables for the response to the reviewer'-table s7).

4. RESULTS- CRUDE–use of an SSRI in 3 months prior to conception – 1.62 (1.33-1.96) – and gender analysis seems to show this is mainly effecting the boys, at least in the short term initially – Boys: 1.42 (0.89-2.26) vs Girls: 0.67 (0.17-2.69).

Response: Considering the limited number in the subgroups of girls (only 2 cases in girls whose father use SSRIs only during the last three months before conception), this observed risk (crude HR=0.67) might not reflect the true effect of paternal SSRIs exposure on girls. Besides, when we looked into the results of main analysis rather than the former-user analysis in boys and girls, the crude HR in boys and girls is 1.61(95%CI: 1.30-2.00) and 1.63(95%CI: 1.05-2.53), respectively, both of which was similar to the HR in all children-1.62 (95%CI: 1.33-1.96). Thus, the evidence to support a stronger effect on boys is limited.

Former vs LT users vs ST users (adjusted) – 1.54 (1.21-1.94) vs 1.32 (1.02-1.72) vs 1.17 (0.75-1.82) – these are the results after adjustment for 9 potential confounders.

Response: Thank you for your suggestions. We have made corrections in the RESULTS section accordingly.

5. DISCUSSION-Authors open their Discussion with ‘we observed an increased crude risk of ASD in the offspring following paternal use of SSRIs during the last 3 months prior to conception’ however claim that following adjustment for confounders that this risk disappears, which is misleading. Even after the adjustment of NINE ‘possible’ confounders the results still show an increased risk.

Response: We are sorry for the misleading texts. However, we did not mean that the risk disappeared after the following adjustment for confounders. Instead, we stated ‘the risk attenuated after adjusting for a number of potential confounders’ in the original version, and we did not deny the existing increased risk after adjustment.

‘Risk persists among former users but not current’ – results showed an increase risk for both but higher in former. Also labeling ‘current’ as ‘current SHORT TERM’ user might be more accurate.

Response: As the results shown in table2, in the main analysis, the adjusted risk was 1.43(1.18-1.74). However, in the former-user analysis (i.e., re-categorized the exposed children into three subgroups: former users, current users, both former and current users), the increased risk persisted in former users (aHR=1.54, 95%CI: 1.21-1.94). As for the current users, the risk attenuated closely to 1 and lost significance (aHR=1.17, 95%CI: 0.75-1.82). That was why we said ‘Risk persists among former users but not current’.

We agree with you on that the current users are all SHORT TERM users compared to those former users, so we have adopted your advice and added a definition about current users in METHOD section to make it more accurate.

Comment: Authors state ‘When we looked into those children without paternal affective disorder before birth of children, similar patterns of association to the main analyses were found. Thus, in addition to affective disorders, other indications related to SSRIs use may also contribute to the observed associations.’ The analysis showed a higher risk in those children born to fathers without a mood disorder (1.57: 1.27-1.93) compared to those with a diagnosis of a mood disorder: 1.17: 0.65-2.09). This does not support the argument that a diagnosis of a mood disorder itself is a significant risk factor, at least in men.

Response: The purpose of this stratified analysis was to distinguish the effect of paternal SSRIs use from that of the main indication (i.e., affective disorders) for SSRIs treatment. In the present study, when we looked into those children born to fathers with affective disorders and examined the isolated effect of paternal SSRIs use in this subgroup (since both of the exposed group and the reference group were with paternal affective disorders, the ASD risk could solely be attributable to paternal SSRIs use), there was no association between paternal SSRIs use and ASD in children (HR=1.11, 95%CI: 0.65-2.09), which provided an additional evidence that paternal SSRIs use before conception might not be related with the risk of ASD in children.

As for those children born to fathers without affective disorders, the increased risk indicated that other psychiatric diseases related to SSRIs use might be a risk factor for ASD of children. In addition, most fathers might receive SSRIs treatment from their general practitioner and were therefore not registered with a diagnosis of affective disorders in the hospital system. Therefore, it was possible that the increased risk associated with prenatal SSRIs use was partly confounded by paternal affective disorders diagnosed outside a hospital department for which we were not able to adjust. We apologized for the confusing explanations in this part of DISCUSSION, and we have added some explanations both in the METHOD and DISCUSSION section to make it more reasonable (Please review in the revised manuscript-Page9, the last paragraph; Page13, the last paragraph).

6. OVERALL CONCLUSION: 'Our evidence does not support that paternal SSRI use before conception increases the risk of ASD in the offspring, but implies that paternal underlying indications related to SSRI use, or other unmeasured confounding factors may play a role.'

The results show that there IS an increased risk of ASD associated with paternal use of an SSRI and this risk seems more pronounced even after cessation of the drug and over the longer term. The results also show that a diagnosis of an affective disorder does not increase the risk of ASD in offspring but that the risks seem higher in offspring from patients with other indications requiring the use of an SSRI.

Response: As you pointed out, we did observe that there was an increased risk of ASD associated with paternal use of an SSRI. However, SSRIs use usually is an indicator of underlying diseases. We have used several analytic strategies to disentangle the effects of SSRI use from the underlying diseases. We concluded that the observed increased risk of ASD associated with paternal SSRIs use before conception could be a result of confounding by paternal psychopathology, which may be supported by following findings: 1) a significant decline of association after adjustment for paternal psychopathology; 2) the similarly increased ASD risk observed in former long-term users rather than in current short-term users; 3) null association in exposed children with paternal affective disorders, which may indicate no effects of SSRIs use; and 4) similar ASD risk among exposed and unexposed siblings. Based on the evidence, we assumed that paternal psychiatric illness rather than SSRIs exposure might be associated with the increased risk of ASD. Besides, as we explained in the last response, the results of stratified analyses actually described the effect of paternal SSRIs use but not paternal affective disorders. Some expressions may be misleading in the manuscript; therefore, we revised the related sentences and the OVERALL CONCLUSION to make them more clear (Please review in the revised manuscript-Page16, the first paragraph).

It would be helpful if these results were broken out by drug where possible. The assumption is that all serotonin reuptake inhibiting drugs produce changes in sperm count and function but any evidence on this would be useful for formulating further investigations. The pattern points to effects on sperm count and function rather than the availability of the drug in seminal fluid.

Response: Thank you for your suggestion. We have done the breakdown analyses to look at the effect of different subtypes of SSRIs, and the results did not change remarkably in whole as the main analyses. However, rare numbers of cases for several drugs may not provide useful information (please see our response to your third comment and table s3-s6 in supplements). Although we made the conclusion that paternal drug use before conception might not impact much on the neurodevelopment of the fetus, however, it would be useful and valuable to formulate a new study to test the effects of SSRIs on sperm count and function, which would be a basis for further studies on paternal SSRIs use and offspring health.

Reviewer: 2

The manuscript reported non-significant/negative results in finding any role of paternal SSRI use in developing ASD in children. It cited only references 11-13 to mention about the seemingly [still not enough to confirm] significant roles of maternal SSRI use in developing ASD. It seems that the current study design was too limited to analyse and contrast the possible roles of paternal and maternal SSRI use. Without the data contrast (positive in maternal roles and negative in paternal roles) in the same study, it would not be convincing to me that the study is highly significant in design and sensitive in methods.

Response: Thanks a lot for your comments. We have now incorporated newest findings from recent studies in our discussion. It has been an important research topic on the association between exposure to maternal SSRI use during pregnancy and offspring neurological / psychiatric outcomes. While the exact role of maternal SSRIs use in ASD of children is still not clear, less is known about effects of paternal SSRI medication. We used the unique data source to examine our hypothesis. Although there is a number of limitations, we still believe the strengths of this study can probably provide the best available evidence on the association between paternal SSRI use in prenatal period and offspring ASD. Firstly, to disentangle the effect of maternal medication from paternal medication, we adjusted the maternal SSRIs use in regression model, and also restricted the analyses to children whose mothers neither received antidepressant medication during pregnancy nor had affective disorders before child birth. Secondly, this was a large population-based cohort study with 669,922 child-father pairs, which could provide the best available power to examine our research questions. The information on exposure to SSRIs was based on a national registry, which eliminated the risk of recall bias. The information on ASD diagnosis was obtained independently of exposure measurement, which could also reduce the information bias. Furthermore, the good data on covariates enabled us to adjust for a number of potential confounders including socio-demographic factors and parental psychiatric history. Thirdly, we also used several analytic strategies to account for confounding by indication as much as we could: (1) regression adjustment; (2) negative controls (i.e., former-users analyses); (3) stratified analyses according to paternal history of affective disorders; and (4) sibling analyses.

VERSION 2 – REVIEW

REVIEWER	Professor David Healy Department of Psychological Medicine Betsi Cadwaladr University Health Board/ Bangor University, UK
REVIEW RETURNED	26-May-2017

GENERAL COMMENTS	The paper is much improved and although we believe the data suggests a much higher risk of ASD in children whose fathers had taken an SSRI than is stressed by the authors, this is still an important paper to publish and will contribute highly to the existing literature on this topic. The addition of a breakdown of the results by individual drug is particularly valuable and the results look comparable to the data available in children born to mothers taking an SSRI.
---

REVIEWER	Siu-wai Leung University of Macau, China
REVIEW RETURNED	12-Jun-2017

GENERAL COMMENTS	The authors responded to my previous concerns by three major arguments for supporting the strengths of evidence; however, the manuscript did not argue for the evidence strength. I am aware of some revision made to the ethics approval statements, which caught my attention. I am not sure whether any ethics approval is unnecessary for this study. It would be nice if the ethics committee can take a look at this study and confirm about the requirement.
--

REVIEWER	Nick de Klerk Telethon Kids Institute, University of Western Australia, Australia
REVIEW RETURNED	31-Aug-2017

GENERAL COMMENTS	This is a nice paper but I have two reservations: I think the coding of the time windows needs to be simplified into any or no exposure in each period: last 3 months, 3 months-1 year, and 1-2 years. Then include all 3 variables in each model and possibly their interactions. Aside from leading to more easily interpretable effect estimates, this would I think remove some of the confusion (in my mind anyway) as to why in all the tables, the numbers of 'any use during the last 3 months' is not equal to the sum of 'only use in the last 3 months' and 'use before and during the last 3 months'. In any case I think those categories need clarification. In the family study (Table 4), it's not clear in the Methods how (or if) the within family similarities have been taken account of, or how the 'unmeasured family-related confounding factors' have been controlled for – unless in the stratified Cox regression the strata were families (which would mean all families with no ASD would have been ignored). Which strata were used there and in the previous analyses need to be clarified. As a minor point, I'd also like to see some justification for using Cox regression, which usually gives more weight to earlier diagnosed cases, and there is some evidence that ASD occurs at or before birth and that the age of diagnosis depends on many other external and extraneous factors (feel free to disagree).
--

VERSION 2 – AUTHOR RESPONSE

Reviewer: 1

- The paper is much improved and although we believe the data suggests a much higher risk of ASD in children whose fathers had taken an SSRI than is stressed by the authors, this is still an important paper to publish and will contribute highly to the existing literature on this topic. The addition of a breakdown of the results by individual drug is particularly valuable and the results look comparable to the data available in children born to mothers taking an SSRI.

Response: Thank you very much for your comment.

Reviewer: 2

- The authors responded to my previous concerns by three major arguments for supporting the strengths of evidence; however, the manuscript did not argue for the evidence strength.

Response: We thank the reviewer for the suggestions. We have now added the arguments for supporting the strengths of evidence in the revised manuscript. (The third and fourth paragraph in DISCUSSTION section)

- I am aware of some revision made to the ethics approval statements, which caught my attention. I am not sure whether any ethics approval is unnecessary for this study. It would be nice if the ethics committee can take a look at this study and confirm about the requirement.

Response: Thank you for your suggestion, our study has already been approved by the Danish Data Protection Agency (Document No. 2013-41-2569). In our study, secondary data from national registers were retrieved using encrypted identification numbers and all analyses were performed at a secure platform at Statistics Denmark without access to personal identification numbers of the participants. Approval by an institutional review board and informed consent are not always required for registry-based research in Denmark.

Reviewer: 3 (statistical review)

This is a nice paper but I have two reservations:

- I think the coding of the time windows needs to be simplified into any or no exposure in each period: last 3 months, 3 months-1 year, and 1-2 years. Then include all 3 variables in each model and possibly their interactions. Aside from leading to more easily interpretable effect estimates, this would I think remove some of the confusion (in my mind anyway) as to why in all the tables, the numbers of 'any use during the last 3 months' is not equal to the sum of 'only use in the last 3 months' and 'use before and during the last 3 months'. In any case I think those categories need clarification.

Response: We apologized for the vague message. We first defined the susceptible exposure time window as the last 3 months before conception in the main analysis. Then we extended the exposure time window to 1 year before conception in the sub-analyses, in order to examine the effect of SSRIs use in the previous 9 months, but not during the defined 3-month susceptible period. This so-called negative-control strategy allowed us to identify a group of men who used SSRIs only during the period from 12 months to 3 months before conception (i.e., former users) and the other two groups of men who used SSRIs during the 3-month susceptible period: those who used SSRIs only during the last 3 months before conception (current users) and those who used SSRIs both before and during the last 3 months before conception (both former and current users). The reason why the case number in the main analysis (N0=104) was larger than the sum of 'current users' (N1=20) and 'both former and current users' (N2=57) was that, N1 and N2 was only limited in the '12 month' window, however, N0 was observed in a wider time axis (i.e., from more than 12 months to conception).

We have considered this suggestion, but changing the time windows into three periods (last 3 months, 3 months-1 year, and 1-2 years) would exclude those fathers who continuously used SSRIs at two or three of the periods in the analyses leading to loss of sample size. We hereby retained the original categories. However, we agree that the original table was difficult to follow. We revised the related tables by adding one explanatory row ('Sub-analysis: Paternal SSRIs use during the last 1 year before conception') and hope they are more readable now.

- In the family study (Table 4), it's not clear in the Methods how (or if) the within family similarities have been taken account of, or how the 'unmeasured family-related confounding factors' have been controlled for – unless in the stratified Cox regression the strata were families (which would mean all families with no ASD would have been ignored). Which strata were used there and in the previous analyses need to be clarified.

Response: We apologize for the vague description of the sibling study. We conducted the sibling study based on matched sets of exposure-discordant siblings, in which there was at least one child with paternal SSRIs preconception exposure and one child without exposure. Using stratified Cox proportional-hazards regression models in which the stratum was families, we compared the ASD risk in exposed siblings to that in unexposed siblings. Sibling-pairs discordant for the studied exposure (i.e., paternal SSRIs use before conception), instead of the outcome (ASD in children), were selected. Therefore, those families with siblings having consistent exposure status have been ignored. Besides, the proportional hazard assumption was evaluated for all variables included in the adjusted Cox models by comparing estimated log-minus-log survival curves.

To clarify, we have revised the related contents about sibling analysis and Cox proportional-hazards regression model in the Statistical Analysis section.

- As a minor point, I'd also like to see some justification for using Cox regression, which usually gives more weight to earlier diagnosed cases, and there is some evidence that ASD occurs at or before birth and that the age of diagnosis depends on many other external and extraneous factors (feel free to disagree).

Response: Before conducting the Cox regression analysis, the proportional hazard assumption has already been checked for all variables included in the adjusted model by comparing estimated log-minus-log survival curves. We also used logistic regression model as sensitivity analysis which led to similar results. For example, by using logistic regression, the fully adjusted OR for ASD among exposed children was 1.53 (95% CI: 1.25-1.86). When extending the exposure window to the last one year before conception, the fully adjusted OR for ASD in former users and current users was 1.65 (95%CI: 1.30-2.09) and 1.23 (95%CI: 0.79-1.92), respectively. The result of logistic regression showed the same pattern as observed in Cox regression.

It is true that there is more weight to earlier diagnosed cases in cox regression. We agree that ASD may occur at or before birth, and diagnosis depends on many other external and extraneous factors which may affect the causal inference of Cox regression. However, if other external and extraneous factors are not differentially distributed in exposed and unexposed group, our results will not be biased. Or those other external and extraneous factors can be well adjusted in the model, our results will not be biased either. We have data on some of these variables, and we hope that adjusting for them could reduce the bias to some extent. For other variables on which we do not have data, our ability to control their confounding effect is limited. We have now added this as one limitation in the DISCUSSION section: 'Fourthly, the age of ASD diagnosis which was used as time event in Cox regression models might be affected by external and extraneous factors. If these factors are differentially distributed in exposed and unexposed group, the actual associations may be biased. We have adjusted for some factors which may influence age of diagnosis to reduce the bias to some extent. However, we could not rule out the confounding effects of unmeasured factors, which is a limitation of the study.'

Thank you again to the editors and reviewers for all the valuable comments above which have been very helpful in improving our manuscript significantly.

VERSION 3 – REVIEW

REVIEWER	Nick de Klerk Telethon Kids Institute, University of Western Australia
REVIEW RETURNED	16-Oct-2017
GENERAL COMMENTS	This is much improved and all my queries have been clarified. I think though that the authors may have missed my point about the re-coding of exposure, in that no-one would be excluded, as they were not mutually exclusive categories, just 3 separate 0/1 variables, where the father would have a 1 or 0 in all 3. It's not worth re-doing at this stage, as I doubt if it would affect any conclusions.

VERSION 3 – AUTHOR RESPONSE

Reviewer: 3

-This is much improved and all my queries have been clarified.

I think though that the authors may have missed my point about the re-coding of exposure, in that no-one would be excluded, as they were not mutually exclusive categories, just 3 separate 0/1 variables, where the father would have a 1 or 0 in all 3. It's not worth re-doing at this stage, as I doubt if it would affect any conclusions.

Response: Thank you for your comment.